# Five Models for Five Modalities: Open-Vocabulary Segmentation in Medical Imaging

Lavsen Dahal[1][0000−0002−8991−759X], Yubraj Bhandari[1][0009−0004−7279−4097], William Paul Segars[1][0000−0003−3687−5733], and Joseph Lo[1][0000−0002−9540−5072]

Duke University, Durham NC, USA
{lavsen.dahal}@duke.edu

**Abstract.** We present a multimodal approach to open-vocabulary segmentation in medical imaging by training five modality-specific models using a unified architecture based on the SAT model. Each model is tailored to a specific imaging modality—CT, MRI, Ultrasound, Microscopy, and PET, while maintaining architectural consistency to ensure comparability and generalizability. To address the challenge of limited data availability, particularly in modalities like Ultrasound and Microscopy, we implement distinct sampling strategies designed to maximize anatomical and pathological diversity across training cases.
We aim to evaluate the effectiveness of open-vocabulary segmentation across diverse medical imaging modalities using consistent text prompts and unified label representations. For CT, MRI, and Ultrasound, performance is reported using Dice Similarity Coefficient (DSC) and Normalized Surface Dice (NSD), while for Microscopy and PET, we follow challenge-specific guidelines and report F1 scores. On the official test set, the models achieved: CT (DSC: 0.2884, NSD: 0.2114), MRI (DSC: 0.1644, NSD: 0.1474), Microscopy (F1: 0.4502), and PET (F1: 0.0728). These results demonstrate the viability of modality-specific training within an open-vocabulary framework and provide a foundation for further improvements.

**Keywords:** open vocabulary segmentation · CT · MRI .

## 1 Introduction

Supervised medical image segmentation has traditionally relied on fixed-class models trained with dense annotations. Large-scale efforts such as TotalSegmentator [16] and DukeSeg [1] exemplify this approach by enabling high-accuracy segmentation across 100+ predefined anatomical structures in Computed Tomography (CT) scans. While these models are robust within their respective label sets, they inherently lack the flexibility to handle unseen categories or user-defined prompts. This limitation has driven increasing interest in open-vocabulary segmentation, which enables segmentation tasks via free-text descriptions [22]. Foundation models such as Segment Anything Model (SAM)[8]

and SAM2[14] have demonstrated impressive segmentation capabilities through user-interactive prompts in natural image domains. However, these interactive foundation models inherently lack support for text-based guidance. Extending open-vocabulary segmentation principles from natural images [9], [18], [17] to medical images introduces unique complexities. Medical datasets frequently suffer from limited annotated data, requiring innovative sampling strategies for robust model training. Moreover, the computational demands of open-vocabulary models challenge their feasibility on large, high-resolution medical volumes.

Further complicating medical image segmentation is the intrinsic diversity of medical imaging modalities. CT scans are volumetric with relatively coarse textures, whereas microscopic images reveal cellular-level details at substantially higher resolutions, involving vastly different texture patterns. Such variability, in dimensionality, resolution, and texture—makes it particularly challenging to develop a universally effective segmentation model. To overcome these hurdles, recent interactive medical segmentation methods such as SegVol [2], SAM-Med3D [15], VISTA3D [4], and nnInteractive [3] leverage user interactions for refinement but do not support open-ended text prompts. Text-guided segmentation approaches, by contrast, explicitly leverage natural language. BioMed-Parse [20] initiated this strategy for 2D biomedical images, while CAT [5] and SAT [21] have successfully extended text-guided open-vocabulary segmentation to 3D medical modalities.

Given the substantial diversity in medical imaging modalities—such as Computed Tomography (CT), Magnetic Resonance Imaging (MRI), Ultrasound, Microscopy, and Positron Emission Tomography (PET)—developing a single universal segmentation model is highly challenging. Unlike natural images, these modalities vary significantly in dimensionality, texture, and resolution. Crucially, in clinical practice, clinicians inherently know the imaging modality being utilized, and this modality-specific information can be strategically leveraged to enhance segmentation performance. Therefore, rather than training one universal model, we propose training separate modality-specific models, explicitly tailored to the unique characteristics of each imaging type. Building upon the recently proposed SAT model [21], which has demonstrated strong performance in text-guided 3D medical segmentation, we develop five distinct SAT-based models [21], each fine-tuned with targeted sampling strategies optimized for their respective modality.

In this challenge, we leverage the provided text prompt to automatically infer the imaging modality. Once the modality is identified from the input prompt and image, the corresponding modality-specific model is dynamically selected and executed. This ensures optimized segmentation performance by automatically routing the task to the best-suited modality-specific SAT model, streamlining the segmentation process across diverse medical imaging scenarios.

## 2   Method

In our approach, we develop five independent models, one for each imaging modality (CT, MRI, PET, Ultrasound, and Microscopy), all based on the same SAT architecture [21] but trained with modality-specific data and weights.

### 2.1   Network Architecture

Figure 1 illustrates our modality-specific adaptation of the SAT architecture. Each model shares the same underlying structure but is independently trained on data from a specific imaging modality, allowing the architecture to specialize in the visual characteristics unique to that domain. During inference, a prompt parser identifies the modality from the input text, enabling automatic selection of the corresponding SAT model to perform segmentation. This design preserves architectural consistency while enabling tailored performance across diverse medical imaging modalities.

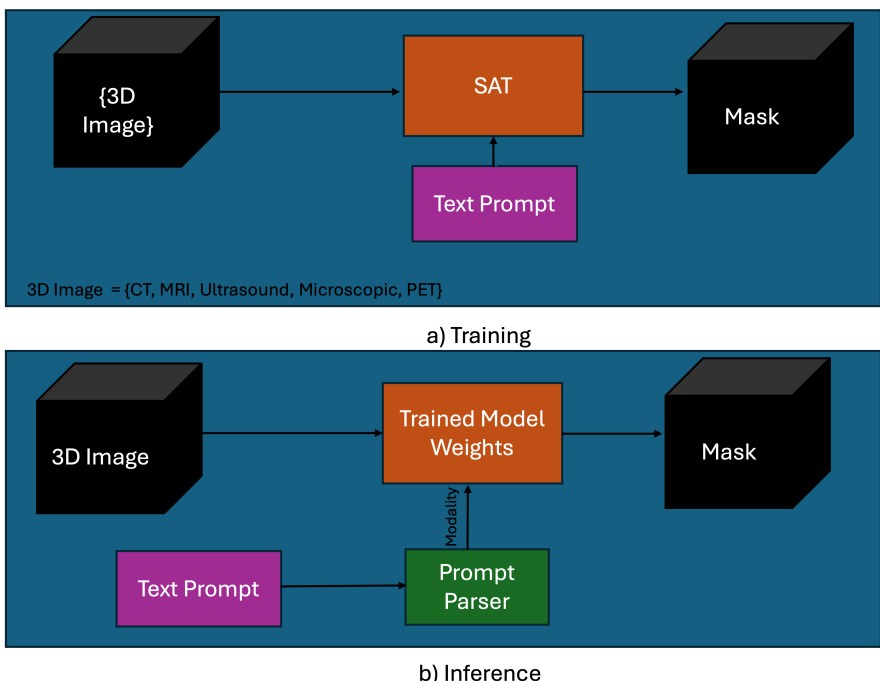

**Fig. 1.** (a) During training, we independently train five modality-specific models, one each for CT, MRI, PET, Ultrasound, and Microscopy, using the same SAT architecture, with separate weights for each modality.(b) During inference, a prompt parser module analyzes the input text to infer the imaging modality. Based on the inferred modality, the corresponding SAT model is selected and used to generate the segmentation mask.

## 2.2   Prompt Encoder and Decoder

**Encoder**  We are using the Text Encoder of SAT  [21] that uses a BERT-based transformer trained on biomedical texts, further enhanced via contrastive learning using anatomical definitions and visual examples. It outputs a knowledge-rich embedding of the medical term.

**Decoder**  A transformer query decoder refines the text embedding by attending to image features. The final segmentation mask is generated by computing similarity between the refined text embedding and image features.

## 2.3   Loss Function

The SAT model uses a combination of Dice loss and Binary Cross-Entropy (BCE) loss  [10]. Dice loss handles class imbalance by focusing on overlap between predicted and ground truth masks, while BCE ensures pixel-level accuracy.

## 2.4   Coreset selection strategy

For the Coreset Track, we were restricted to using only 10% of the full training dataset. To construct a representative and diverse subset, we designed a sampling strategy that ensures broad coverage across modalities and datasets while favoring samples rich in segmentation labels.

Our sampling process adhered to several key constraints: (1) the final subset must be approximately 8.2% of the total dataset size, (2) each dataset must contribute at least five samples to maintain diversity across sources, and (3) each imaging modality must be represented by at least fifty samples to preserve modality balance.

We first filtered out corrupted or invalid data and computed label presence for all usable files. Sampling was performed in two phases. In the first phase, we enforced the per-dataset and per-modality minimums through weighted random selection, where samples with more labeled structures were more likely to be chosen. In the second phase, we filled the remaining quota with globally sampled files, again guided by label richness. To ensure consistency and reproducibility, a fixed random seed was used throughout. The resulting subset was saved in a structured JSONL format and used to train our model.

## 2.5   Prompt-Based Modality Router (No Text Conditioning)

*Scope.* The prompt is used *only* to select the modality-specific SAT model at inference. We do *not* condition the decoder on text, and we do *not* parse organ names to alter outputs. Each modality-specific SAT predicts its predefined label set unchanged.

*Procedure.*

1. **Normalize & tokenize:** lowercase the prompt and tokenize with word boundaries to avoid spurious matches.
2. **Keyword detection:** search for modality keywords
   {`ct`, `mri`, `ultrasound`, `microscopy`, `pet`}.
3. **Model selection:** if a keyword is detected, load that modality's SAT weights and run segmentation; otherwise, **fallback to CT** (challenge default).

*Notes.* This router is rule-based (no learned parameters) and *does not* alter the segmentation network or its label heads. Any per-structure visualization uses the model's standard outputs; prompts are not used to gate or condition class predictions. Because the challenge enforced a strict per-case runtime limit, we intentionally kept the prompt parser lightweight and did not add extra complexity such as a separate image-based modality detector or a learned router.

## 3  Experiments

### 3.1  Dataset and evaluation metrics

The development set is an extension of the CVPR 2024 MedSAM on Laptop Challenge [12], including more 3D cases from public datasets[1] and covering commonly used 3D modalities, such as Computed Tomography (CT), Magnetic Resonance Imaging (MRI), Positron Emission Tomography (PET), Ultrasound, and Microscopy images. The hidden testing set is created by a community effort where all the cases are unpublished. The annotations are either provided by the data contributors or annotated by the challenge organizer with 3D Slicer [7] and MedSAM2 [13]. In addition to using all training cases, the challenge contains a coreset track, where participants can select 10% of the total training cases for model development.

The text-guided segmentation task contains both semantic segmentation and instance segmentation. For the semantic segmentation task, the evaluation metrics include Dice Similarity Coefficient (DSC) and Normalized Surface Distance (NSD) to evaluate the segmentation region overlap and boundary distance, respectively. For the instance segmentation task, we computed the F1 score at an overlapping threshold of 0.5 and DSC scores for true positives. In addition, the algorithm runtime will be limited to 60 seconds per class. Exceeding this limit will lead to all DSC and NSD metrics being set to 0 for that test case.

### 3.2  Implementation details

**Preprocessing** Following the practice in MedSAM [11], all images were processed to npz format with an intensity range of $[0, 255]$. Specifically, for CT images, we initially normalized the Hounsfield units using typical window width

---

[1] A complete list is available at https://medsam-datasetlist.github.io/

and level values: soft tissues (W:400, L:40), lung (W:1500, L:-160), brain (W:80, L:40), and bone (W:1800, L:400). Subsequently, the intensity values were rescaled to the range of $[0, 255]$. For other images, we clipped the intensity values to the range between the 0.5th and 99.5th percentiles before rescaling them to the range of $[0, 255]$. If the original intensity range is already in $[0, 255]$, no preprocessing was applied.

**Environment settings** The development environments and requirements are presented in Table 1.

**Table 1.** Development environments and requirements.

| System | Ubuntu 20 |
| --- | --- |
| CPU | 128 AMD EPYC 7000 series |
| RAM | 1024 GB |
| GPU (number and type) | Four NVIDIA RTX A6000 48G |
| CUDA version | 12.2 |
| Programming language | Python 3.11.11 |
| Deep learning framework | torch 2.6.0 |

**Data Augmentation** We applied a diverse set of data augmentations, inspired by the nnU-Net framework [6], to improve model generalization and robustness. These included geometric (rotation, scaling, mirroring), intensity (contrast, brightness, gamma, noise, blur), and resolution-based transformations. Augmentations were applied probabilistically during training to simulate real-world variability in imaging conditions and anatomical presentations.

**Table 2.** Training protocols.

| Pre-trained Model | SAT Text Encoder |
| --- | --- |
| Batch size | 1 |
| Patch size | 288×288×96 |
| Total iterations | 50000 |
| Optimizer | Adam |
| Initial learning rate (lr) | 1e-4 and 1e-5 |
| Lr decay schedule | cosine annealing |
| Loss function | BCE and Dice |
| Number of model parameters | 220.9M[2] |

# 4   Results and discussion

**Table 3.** Quantitative evaluation results of the **validation set** on the **coreset track**. Our proposed method, denoted as **SAT-{modality}**, trains a separate model for each imaging modality to better handle modality-specific characteristics. Microscopy metrics are unavailable for some cases due to evaluation errors.

| Modality | Method | Sematic Segmentation | | Instance Segmentation | |
|---|---|---|---|---|---|
| | | DSC | NSD | F1 | DSC TP |
| | CAT | **0.7443** | 0.7444 | 0.2575 | 0.3148 |
| CT | SAT | 0.7028 | 0.7058 | 0.1128 | 0.1534 |
| | SAT-CT(Ours) | 0.3280 | 0.3043 | 0.04 | 0.12 |
| | CAT | **0.4831** | 0.5472 | 0.1498 | 0.3127 |
| MRI | SAT | 0.475 | 0.551 | 0.0575 | 0.175 |
| | SAT-MRI(Ours) | 0.305 | 0.349 | 0.0162 | 0.106 |
| | CAT | ... | ... | 0.043 | 0.3224 |
| Microscopy | SAT | ... | ... | 0.1741 | 0.5802 |
| | SAT-Microscopy(Ours) | ... | ... | **0.5825** | **0.8254** |
| | CAT | ... | ... | 0.1127 | 0.3382 |
| PET | SAT | ... | ... | 0.2736 | 0.7326 |
| | SAT-PET(Ours) | ... | ... | **0.3105** | **0.7352** |
| | CAT | **0.818** | 0.812 | ... | ... |
| Ultrasound | SAT | 0.7611 | 0.7268 | ... | ... |
| | SAT-Ultrasound(Ours) | 0.7656 | 0.7485 | ... | ... |

## 4.1   Quantitative results on validation and test set

Table 3 and Table 4 summarizes our modality-specific adaptation of SAT along-side the baseline SAT and CAT methods. Because we trained our models for far fewer iterations than the baselines, results remain below the performance of the fully trained SAT and CAT on CT and MR modality. However, for modalities with smaller datasets, namely microscopic images, ultrasound, and PET, the limited training still allowed the model to see every example, leading to strong segmentation performance. In fact, our adaptation outperforms both SAT and CAT on PET and microscopic image segmentation, as shown in the Table 3.

## 4.2   Qualitative results on validation set

The qualitative results show that our open-vocabulary model can achieve strong performance on well-represented structures, e.g., lungs in CT, the prostate on T2-weighted MRI, and cardiac chambers in ultrasound—yielding high Dice scores as shown on Figure 2 and Figure 3. At the same time, it entirely failed to delineate branching airway structures in CT and to segment brain tumors on MRI, both of which pose more complex anatomical and contextual challenges that the model

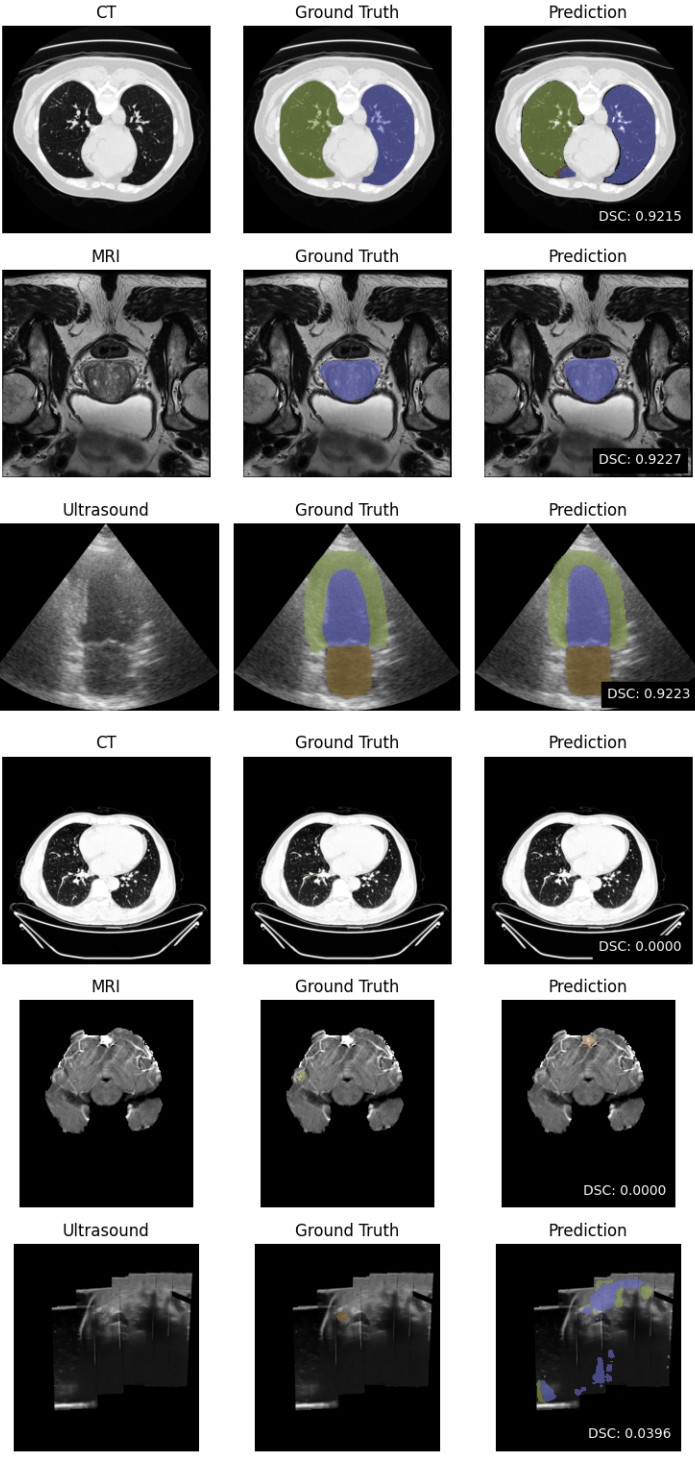

**Fig. 2.** Segmentation results for CT, MRI, and ultrasound modalities, showing both the best- and worst-performing cases (by Dice score) in each modality. The first three rows represent the best-performing cases, while the last three rows depict the worst-performing cases. In each row, the first column displays the original image slice; the second overlays the ground-truth segmentation; and the third overlays the model's prediction with the corresponding Dice similarity coefficient (DSC) annotated.

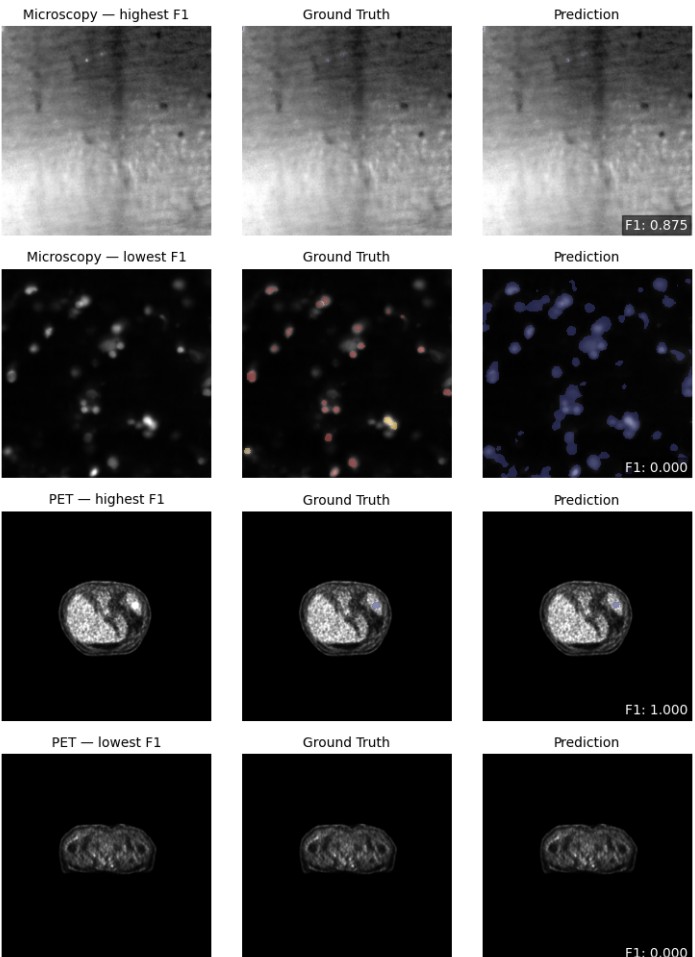

**Fig. 3.** Segmentation results for PET and Microscopy modality, showing both the best- and worst-performing cases (by F1 score) in each modality. The top and bottom two rows represent the best and worst performing cases for the Microscopy modality and PET modality respectively. In each row, the first column displays the original image slice; the second overlays the ground-truth segmentation; and the third overlays the model's prediction with the corresponding F1 score annotated.

**Table 4.** Quantitative evaluation results of the **test set** on the **coreset track**. Our proposed method, denoted as **SAT-{modality}**, trains a separate model for each imaging modality to better handle modality-specific characteristics.

| Modality | Method | Sematic Segmentation | | Instance Segmentation | |
|---|---|---|---|---|---|
| | | DSC | NSD | F1 | DSC TP |
| CT | CAT | 0.3689 | 0.3500 | . . . | . . . |
| | SAT | **0.4314** | 0.3824 | . . . | . . . |
| | SAT-CT(Ours) | 0.2884 | 0.2114 | . . . | . . . |
| MRI | CAT | 0.2667 | 0.2701 | 0.0313 | 0.0398 |
| | SAT | **0.3368** | 0.3314 | 0.0884 | 0.1817 |
| | SAT-MRI(Ours) | 0.1644 | 0.1474 | 0.0154 | 0.0810 |
| Microscopy | CAT | . . . | . . . | 0.0110 | 0.3377 |
| | SAT | . . . | . . . | 0.3116 | 0.7609 |
| | SAT-Microscopy(Ours) | . . . | . . . | **0.4502** | **0.7995** |
| PET | CAT | . . . | . . . | 0.0000 | 0.0000 |
| | SAT | . . . | . . . | **0.0886** | **0.2692** |
| | SAT-PET(Ours) | . . . | . . . | 0.0728 | 0.1808 |

couldn't learn in the limited training time. Likewise, limb segmentation in ultrasound was unsuccessful, reflecting the lack of similar examples in the training set. Altogether, these findings underscore the difficulty of open-vocabulary segmentation in medical imaging, where diverse modalities and intricate anatomy demand richer contextual knowledge than what models typically acquire from natural-image datasets.

### 4.3    Results on final testing set

We participated in the Coreset Challenge, which was conducted on a subset of the full dataset to evaluate segmentation performance under limited data conditions. Our method, submitted under the team name **cvit**, achieved the **4**[th] position in the final rankings. The official results released by the challenge organizers are summarized in Table 5.

**Table 5.** Final testing results of all participating teams. Our team, **cvit**, achieved the **4**[th] position on the final testing set.

| Rank | Team | Avg DSC | Avg NSD | Avg F1 | Avg DSC_TP |
|---|---|---|---|---|---|
| 1 | mirthai-lab | 0.402 | 0.3374 | 0.0283 | 0.082 |
| 2 | zen | 0.3457 | 0.3097 | 0.0533 | 0.1806 |
| 3 | hanglok | 0.3273 | 0.3175 | 0.0270 | 0.0412 |
| 4 | **cvit** | **0.2378** | **0.1853** | **0.0314** | **0.108** |
| 5 | imiphdu | 0.0803 | 0.0558 | 0.0094 | 0.0858 |
| 6 | deepseg | 0.0126 | 0.0147 | 0.0030 | 0.0076 |

### 4.4  Limitation and future work

We trained for approximately 10000 iterations. This limited schedule disproportionately affected **CT** and **MR**, which have larger data volume and tend to benefit from longer training and modest modality-specific tuning.

A major error source was **orientation heterogeneity** in the released `.npz` volumes: orientation metadata was not provided, preventing reliable normalization to a common convention (e.g., RAS/LPS). Under these conditions, standard spatial augmentations (flips/rotations) can inadvertently *mix left/right anatomy*, degrading structures with sidedness (e.g., left/right kidney, ribs).

**Future work** will: (i) add a deterministic orientation harmonization step; (ii) employ flip-aware labeling or sidedness-preserving augmentations; and (iii) incorporate left/right consistency checks to guard against such failures.

Our submission uses a **prompt-only, rule-based router** for modality selection at inference: keyword detection for {`ct`, `mri`, `ultrasound`, `microscopy`, `pet`}, with a **CT fallback** when the prompt lacks explicit modality cues (challenge default). This router performs *only* model selection and does not alter the segmentation network; no text-conditioned decoding is used in this work. **Future work** will strengthen routing by replacing fixed fallback with a lightweight image-based heuristic and a compact CNN modality classifier.

Given substantial heterogeneity across imaging types, we favor **five separate SAT models (one per modality)** over a single model. This enables modality-appropriate choices later (e.g., losses tuned for small structures; diffusion-style heads for tumor/pathology). These directions are **future work** and not part of the reported results.

## 5  Conclusion

In this work, we trained five separate models, each dedicated to a specific imaging modality, rather than relying on a single unified model. We believe this modality-specific strategy is more effective given the significant differences in data characteristics and anatomical structures across modalities.

Preliminary results indicate promising performance on modalities such as microscopy and ultrasound, where the models were able to converge well within the limited training period. However, for modalities like CT and MRI, which typically require more extensive training due to larger datasets and greater variability, convergence was not achieved within the restricted timeframe.

**Acknowledgements**  We thank all the data owners for making the medical images publicly available and CodaLab [19] for hosting the challenge platform. This work was funded by the Center for Virtual Imaging Trials, NIH grants P41EB028744, R01EB001838, and R01CA261457.

**Disclosure of Interests.**  The authors have no competing interests to declare that are relevant to the content of this article.

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

**Table 6.** Checklist Table.

| Requirements | Answer |
| --- | --- |
| A meaningful title | Yes |
| The number of authors ($\leq 6$) | Yes (4) |
| Author affiliations and ORCID | Yes |
| Corresponding author email is presented | Yes |
| Validation scores are presented in the abstract | Yes |
| Introduction includes at least three parts: background, related work, and motivation | Yes |
| A pipeline/network figure is provided | Figure number 1 |
| Pre-processing | Page number 5 |
| Strategies to data augmentation | Page number 5 |
| Strategies to improve model inference | No |
| Post-processing | Page number 4 |
| Environment setting table is provided | Table number 1 |
| Training protocol table is provided | Table number 2 |
| Ablation study | No |
| Efficiency evaluation results are provided | Yes |
| Visualized segmentation example is provided | Figure number 2 and 3 |
| Limitation and future work are presented | Yes |
| Reference format is consistent. | Yes |
| Main text $>=$ 8 pages (not include references and appendix) | Yes |