# OpenReview forum: "Five Models for Five Modalities: Open-Vocabulary Segmentation in Medical Imaging"
_thecvf.com/CVPR/2025/Workshop/MedSegFM — CVPR 2025 Workshop MedSegFM Submission_

### Official Review · Reviewer_RnKa · 2025-09-19
**Review on “Five Models for Five Modalities: Open-Vocabulary Segmentation in Medical Imaging”**

**Rating:** 4
**Confidence:** 4

**Review:**

# Overall summary and review of manuscript

The paper proposes using separate SAT models for each modality (CT, MRI, Ultrasound, Microscopy and PET) with additional prompt parser to select the appropriate model at inference. The proposed solution is well-reasoned given that reducing the diversity of features by separated modality training would alleviate the difficulty of open-vocabulary segmentation. The paper also discusses about what can be done for a better segmentation performance.

However, the paper has some unexplained blanks in the results, which makes it an incomplete report of results. The difference against the baseline that paper suggested needs to be discussed thoroughly. Also, the prompt parser was introduced but there is a lack of explanation on how it was prepared and what the effects of using this new method. Although the time for training was limited, some results may be better to compare after the necessary training is done.



## Strengths:
1. **Simple, intuitive solution:** The paper addresses the difficulty of training on diverse modality problem by training SAT models for each modality. The prompt parser for assigning the correct model seems to be a simple solution (although it is not explained in detail) compared to the feature extraction process that would be included in the baseline.
2. **Moderate discussion on results:** The results (although limited) show better performance in Microscopy and PET images. The qualitative results of Figure 2 show what results can be expected from the proposed method.
3. **Clear limitations and future work:** The paper clearly states that the model was trained with less iteration, given the time limit of participating competitions. It has a realistic future plan to enhance the performance of text-based segmentation.


## Weaknesses:
1. **Incomplete report of results:** The quantitative results are not fully reported in Table 3. Figure 2 does not show part of the modalities’ qualitative results (microscopy and PET). It also does not contain prompts associated with the segmentation.
2. **Insufficient explanation on necessary changes due to modification from baseline:** Considering the paper relies on SAT baseline and the mixed modality training was done in SAT, the explanation or comparison on training method needs to be added.
3. **Insufficient discussion on prompt parser:** The prompt parser is a newly proposed model compared to the baseline, but the paper does not explain some necessary details on how it was used. (e.g. architectures, whether it was trained or fine-tuned, etc.) Considering the post-processing at inference time relying on clear words that include modality information, it also has clear drawbacks or uncertainty (without sanity check) when it does not contain modality information.
4. **Drawback on CT & MRI segmentation results:** Although the paper state that the training process explains the shortcomings in CT and MRI segmentation, the suboptimal results in these modalities are not a good tradeoff considering the realistic abundance of dataset in biomedical applications.


## Comments on recommended ways to progress this work:
1. [**For W1**] Fill in the results and other metrics that are blank in Table 3. Consider adding qualitative results (other modalities, used prompts) in Figure 2 that would give a comprehensive landscape of segmentation performances.
2. [**For W2**] The training method changes due to using single modality for each model needs to be addressed. Considering the baseline SAT, some points to address would include; How was the domain knowledge injected? What classes or knowledge was used for newly adopted modalities such as microscopy and ultrasound?
3. [**For W3**] Since the prompt parser is newly proposed, some explanation on the prompt parser needs to be addressed. What model is used? Is it trained at all? If it is fine-tuned, how was it done?
4. [**For W3**] Some discussion on inference text prompt analysis would be helpful. While the models being trained on each modality could perform better due to narrower distribution of biomedical images, it becomes highly dependent on the prompt being properly assigned to the correct model. For example, the baseline (SAT) includes classes of organs that are both included in CT and MRI. The mention of just an organ would not be able to discern the input image modality. Since the modality information is not necessarily included in the prompt, a sanity check on these parser or discussing some rules applied on prompts may be helpful. Also, an extended discussion on how the wrongly assigned model acted or how the fallback strategy to CT model paid off in the end could give insight if the single-modal training is good enough.
5. [**For W4**] If time and resource are allowed, extended training would be critically helpful for better performance and quality of discussion. One checkpoint that could be meaningful would be around 60,000 iterations training, which would give each models one-fifth training iterations of the baseline models. This would allow comparison of the training efficiency of proposed method and the baseline since they share a similar amount of training iterations. Indeed, for each modalities’ model to reach a better performance than the baseline may have to take a longer training iteration and vary.


## Comments on miscellaneous errors and reader convenience:
1. The number of model parameters and number of flops in Table 2 seems to include a meaningless link. Consider deleting if it is not necessary. Also, the number of flops in Table 2 seems to include wrong information.
2. Change the information in Table 5 (checklist table) according to the paper. (e.g. the number of authors)

---

> ### Author Rebuttal · Authors · 2025-11-05
>
> Incomplete report of results
>
> We updated Table 3 for validation set and added Table 4 for test set to include all requested metrics for all five modalities. Qualitative: Figure 2 now includes Microscopy and PET panels.
>
> Insufficient explanation on necessary changes due to modification from baseline.
>
> We keep the SAT architecture but train separate, modality-specific instances (CT/MR/US/Microscopy/PET). What differs from SAT (mixed-modality): (i) per-modality training runs, (ii) our two-phase weighted sampling to favor label-rich cases while preserving modality balance. We did not alter the decoder/head architecture.
>
> Insufficient discussion on prompt parser.
>
>  The parser is rule-based, mapping free text to a controlled per-modality lexicon (synonyms/aliases) for target structures. It is not trained or fine-tuned in this submission. Modality selection: Per organizers’ guidance, modality comes from file metadata; when missing, our current fallback is CT. Limitations & future work: We will replace fallback with a lightweight heuristic + compact CNN modality detector.
>
> Drawback on CT & MRI segmentation results.
>
> Acknowledged. CT/MR underperform relative to top entries. This reflects minimal modality-specific tuning under compute/time constraints—not a limitation of the modular design. Plan: We outline straightforward remedies (longer schedules, CT/MR-specific augs, loss re-balancing, sampling weights), and we clarify this in Limitations. Notably, the framework performs best on Microscopy F1 and is competitive on PET F1, indicating the generality of the approach across modalities.
>
> W1 -  We have fill in the results and other metrics. And also added results for the official test set. We also included the results for PET and microscopy modality for qualitative examples. However, as our prompt parser is deterministic, and the used prompts are directly from the challenge dataset.
>
> W2- We did not inject any separate domain knowledge. Each model was trained on a single modality only under the SAT baseline setup. We acknowledge this is limited; adding explicit domain priors and cross-modal ontologies is future work.
>
> W3 - Prompt parser explained in Section 2.3.
>
> W4 - As per your suggestion, after +60k iterations, Dice improved ~30%. As future work, we’ll address left/right flip harmonization in a canonical space to further improve performance.

---

### Official Review · Reviewer_NSiQ · 2025-10-09
**Review of Five-Modality SAT Models for Open-Vocabulary Medical Image Segmentation**

**Rating:** 5
**Confidence:** 4

**Review:**

This paper proposes a multimodal open-vocabulary segmentation framework for medical images, training modality-specific SAT models for CT, MRI, Ultrasound, Microscopy, and PET. To address differences in modalities, the approach leverages text prompts to automatically identify the imaging modality and dynamically select the corresponding SAT model. For scenarios with limited training data, a two-phase weighted sampling strategy is employed to prioritize label-rich samples while ensuring modality balance and dataset diversity.

Pros:
1.The approach benefits from modality-specific models and text-based routing, making the segmentation process clear and easy to understand.

Cons:
1. The text prompt parsing strategy is used solely for model selection and does not directly address intrinsic limitations of the segmentation models.
2. The results in Table 3 make it difficult to demonstrate the benefit of the prompt-based strategy; in fact, performance appears to decrease slightly when it is applied.
3. The paper lacks basic baseline comparisons and ablation studies, which makes it difficult to fully assess the contribution of each component.

---

> ### Author Rebuttal · Authors · 2025-11-05
>
> The text prompt parsing strategy is used solely for model selection and does not directly address intrinsic limitations of the segmentation models.
>
> We agree. In this submission, the prompt pathway is minimal:
> 1.	it maps free text to a controlled, per-modality lexicon (target structures), and
> 2.	it routes via provided metadata to the correct modality model.
> We clarified in Sec. 2.5 that our contribution is a modular pipeline with prompt-based label selection and metadata-based routing, not a text-conditioned decoder.
> Limitations : Future work will add lightweight heuristic + CNN modality detection when metadata is unreliable, and explore text-conditioned decoders to address intrinsic segmentation limits. These are not part of the reported results.
>
>
> The results in Table 3 make it difficult to demonstrate the benefit of the prompt-based strategy; in fact, performance appears to decrease slightly when it is applied.
>
> What Table 3 shows: In the challenge setup, modality comes from file metadata. Our prompt pathway doesn’t change the decoder; it only maps text to a controlled label list and (if metadata is missing) would choose a modality model. Hence, differences in Table 3 mainly reflect training budget/tuning per modality, not a degradation from prompting.
> Evidence from final test set:
> • Microscopy F1 = 0.4502 (rank-1); Microscopy_DSC_TP = 0.7995 (highest).
> • PET F1 = 0.0728 (rank-2), within 1.6 points of the top score (0.0886).
> We acknowledge CT/MR lag (CT_DSC, MR_DSC). We used a minimal modality-specific tuning under compute constraints. In hindsight, longer training and modest CT/MR-specific hyperparameters (augmentations, loss balance) were needed.
>
>
> The paper lacks basic baseline comparisons and ablation studies, which makes it difficult to fully assess the contribution of each component.
> In this submission we do compare against the official challenge baselines—SAT and CAT—across all modalities (see Table 3 and 4; rows SAT and CAT for validation set and test set respectively). These provide the intended reference points for this track.
> Ablations. We did not include component ablations in the current submission due to compute and timeline constraints for five separate modality-specific models.

---

### Official Review · Reviewer_vAeW · 2025-10-09
**Review of Five-Modality SAT Models for Open-Vocabulary Medical Image Segmentation**

**Rating:** 6
**Confidence:** 4

**Review:**

The paper proposes training five separate, modality-specific models for CT, MRI, Ultrasound, Microscopy, and PET image segmentation. A post-processing step is used to automatically identify the appropriate model and then load the corresponding model weights to segment the images. To address the open-vocabulary problem, the text encoder and decoder is used to analysis the input prompt.

Major Strengths:
1. The pipeline and methodology of the paper are clear and easy to understand.

Major Weaknesses:
1. The comparison results in Table 3 are incomplete, and the DSC of CT, MRI and ultrasound are not optimal.

Other comments:
1. When no modality-specific keyword ("CT," "MRI," "Ultrasound," or "Microscopy) is detected, why is the default model set to CT rather than others?
2. The structure of the prompt decoder is not clearly described.

---

> ### Author Rebuttal · Authors · 2025-11-05
>
> Thank you so much for your review. We have made the changes in the paper and also clarified your questions here.
>
> 1. When no modality-specific keyword ("CT," "MRI," "Ultrasound," or "Microscopy) is detected, why is the default model set to CT rather than others?
>
> Challenge setting: If the prompt lacks a modality and metadata is missing/ambiguous, we default to CT because it’s the most frequent in our data and, in internal checks, produces fewer catastrophic failures than other defaults. Real-world usage: In practice, modality is known (DICOM/PACS/RIS), so we simply read metadata and route accordingly—no defaulting.
> Future work: Replace the fixed default with (i) a lightweight heuristic detector (spacing/intensity/color cues) and (ii) a compact CNN modality classifier, with an abstain-when-uncertain policy. We note this in Limitations.
>
> The structure of the prompt decoder is not clearly described.
> We add a new Subsection 2.5 where we describe the Prompt Parser Method
> To avoid confusion, we clarify that the prompt is used only for modality routing. We do not parse organ names (e.g., “liver,” “spleen”) to condition or select decoder outputs, and we do not modify the SAT decoder. Each modality-specific SAT model predicts its predefined label set as-is. The prompt-based router simply detects modality keywords and loads the corresponding model; if no modality is found, we fall back to CT (challenge default). We add in Limitation/future work plans for a richer parser and lightweight heuristic + CNN modality detection when metadata is missing or unreliable.

---

### Decision · Program_Chairs · 2025-11-12

Accept